# OpenReview forum: "Copyright-Bench: Agentic Evaluation of Copyright Law Compliance"
_ICML.cc/2026/Conference — ICML 2026 spotlight_

### Official Review · Reviewer_gKwE · 2026-03-12

**Soundness:** 2
**Presentation:** 3
**Significance:** 2
**Originality:** 2
**Overall Recommendation:** 4
**Confidence:** 4

**Summary:**

This paper studies a new challenge arising from the shift from passive large language models to autonomous LLM agents capable of performing real-world commercial tasks.As agents increasingly retrieve and integrate external assets (e.g., images) into deliverables, their actions may lead to legal violations such as copyright infringement, yet current agent benchmarks evaluate only task success or capability, not legal compliance.
To address this gap, the paper introduces Copyright-Bench, a benchmark designed to evaluate copyright compliance in agentic systems. The benchmark simulates realistic commercial tasks in which agents must choose between legally permissible assets and copyright-restricted assets.
The authors evaluate 11 state-of-the-art models, including proprietary systems (GPT-5.x, Claude 4.5, Gemini 3) and open-weight models (Llama-4, Qwen-3VL, DeepSeek-VL)

**Compliance With Llm Reviewing Policy:**

Affirmed.

**Final Justification:**

Thanks for the authors, my concerns have been addressed and I will raise my score.

**Key Questions For Authors:**

- Can the benchmark be expanded to evaluate copyright compliance for text, code, audio, or video?
- How to deal with complex copyright  issues such as licensing conditions.

**Strengths And Weaknesses:**

S：
- The paper highlights legal compliance as an important evaluation criterion for AI agents, which is largely missing from current benchmarks.
- The tasks simulate common real-world workflows (websites, merchandise, presentations), making the evaluation relevant for practical agent use.

W：
- Copyright status is treated as a binary label, while real-world copyright decisions often involve complex issues such as fair use or licensing conditions.
- Lacking diversity. The benchmark only focuses on image selection, leaving out other important areas like text reuse, code licensing, or multimedia assets.
- The dataset size (200 images) limits the diversity of scenarios that can be evaluated.

---

> ### Author Rebuttal · Authors · 2026-03-31
>
> Dear Reviewer gKwE,
>
> We sincerely thank you for your thoughtful review and for recognizing the practical relevance of our work. We are encouraged by your acknowledgment that our tasks accurately simulate real-world workflows and highlight an important, missing dimension in current evaluations. As you noted, the authors strive to analyze a pressing problem within the field of AI alignment. Below, we address your specific questions and concerns, and we will incorporate these clarifications into the final version.
>
> ---
>
> ## 1. Complex Copyright Issues & Licensing Conditions (W1 & Q2)
>
> **Reviewer's point:**
> *Copyright is treated as a binary label, missing the complexity of fair use or nuanced licensing.*
>
> We agree that real-world copyright is highly nuanced. However, as we are the first to evaluate copyright law compliance specifically in autonomous agentic workflows, we deliberately opted for a narrow framing and clear-cut experimental design. Our development of a binary classification (Public Domain/Safe vs. All Rights Reserved/Restricted) was a deliberate choice to isolate the fundamental decision-making signal of the agents. We think we should first ensure agents follow clear legal rules before testing complex fair use.
>
> As stated in **Appendix A (Limitations and Future Work)**, we recognize that real-world compliance is often more granular, involving conditional licenses such as attribution requirements (CC-BY), share-alike clauses (CC-SA), or commercial-use restrictions.
>
> ---
>
> ## 2. Expanding to Other Modalities (Text, Code, Audio, Video) (W2 & Q1)
>
> **Reviewer's point:**
> *Focuses only on images. Can it be expanded?*
>
> **Addressing Q1:**
> Yes, absolutely. The theoretical formalism of Copyright-Bench (Section 3.1) is fundamentally modality-agnostic. The environment relies on an agent reviewing a pool of assets, inspecting their metadata, and making a selection.
>
> We initially focused on images because visual asset integration (e.g., web design, presentation creation) represents one of the most immediate and legally fraught use cases for current multimodal commercial agents. However, the same framework can be extended to:
>
> - **Code:** Evaluating software engineering agents (e.g., in a SWE-bench style environment) on whether they choose to import MIT-licensed libraries versus proprietary/GPL-licensed code when fulfilling a user's coding request.
> - **Text/Audio/Video:** Evaluating agents tasked with compiling research summaries, background music for videos, or stock footage, requiring them to verify usage rights via APIs or file headers.
>
> We will explicitly expand our discussion in Section 3.4 and the Appendix to outline the roadmap for adapting this framework to code and text modalities if we are granted the additional page in camera ready.
>
> ---
>
> ## 3. Dataset Size (W3)
>
> **Reviewer's point:**
> *The dataset size is 200.*
>
> We chose to prioritize strict experimental control over sheer volume. Furthermore, to ensure real-world validity and ethical compliance (in the research study itself), every restricted commercial image used in our experiments had to be legally purchased and licensed from commercial stock libraries, making a tightly controlled, high-quality dataset the most rigorous and feasible approach.
>
> The 200 assets consist of 100 carefully curated pairs (one compliant, one restricted). As detailed in Section 4.3, constructing these pairs required a rigorous, multi-stage pipeline: ensuring high semantic alignment via CLIP, enforcing aesthetic parity via LAION-Aesthetics, and conducting human verification.
>
> Additionally, while the asset pool is 200 images, the benchmark dynamically pairs these across 3 task families, 4 prompt variations (Neutral, IP-Aware, Hurried, Dismissive), and 4 environment configurations (Flat, Hierarchical, Ambiguous, Web). We run 240 evaluation trajectories per configuration, **resulting in thousands of unique agentic reasoning traces**.
>
> That said, we see this initial release as a starting point and thank the reviewer for the suggestion to expand its scope. If accepted, we plan to modestly increase the dataset (e.g., adding some text/audio cases) and continue maintaining and extending the benchmark over time to improve scenario and asset diversity.

---

> > ### Author Rebuttal · Reviewer_gKwE · 2026-04-03
> >
> > Thanks for the authors, my concerns have been addressed and I will raise my score.

---

> > > ### Author Response · Authors · 2026-04-07
> > >
> > > Dear Reviewer gKwE,
> > >
> > > Thank you for your thoughtful review and for recognizing the importance of evaluating legal compliance in agentic systems. We also sincerely appreciate your constructive feedback on scope, dataset design, and real-world applicability.
> > >
> > > **We are grateful that our clarifications addressed your concerns and that you found the revised explanations helpful. Your suggestions have helped us better position the benchmark and clarify its intended scope as a controlled first step toward a broader evaluation framework.**
> > >
> > > We also appreciate your recognition of the practical relevance of our tasks and your support in raising the score. Thank you again for helping us improve the paper.
> > >
> > > Sincerely,
> > >
> > > The Authors

---

### Official Review · Reviewer_TA3Z · 2026-03-13

**Soundness:** 2
**Presentation:** 2
**Significance:** 3
**Originality:** 3
**Overall Recommendation:** 4
**Confidence:** 3

**Summary:**

Authors introduce Copyright-Bench, a benchmark for evaluating whether LLM agents comply with copyright constraints in realistic commercial workflows such as website creation, merchandise design, and content production. Benchmark focuses on cases where agents must choose between lawful and copyrighted assets and varies prompt framing to test the effect of user intent and time pressure. Also, shows that current agents often violate copyright constraints even when lawful alternatives are available and that violation rates increase under pressure.

**Compliance With Llm Reviewing Policy:**

Affirmed.

**Key Questions For Authors:**

- Could you please report task success and compliance together.

- Statistical analysis or failure analysis would be great.

- A joint compliance-vs-utility view would make the benchmark much more informative.

**Limitations:**

I think the paper asks a genuinely important question and introduces a useful benchmark direction. The shift from "can the agent do the task?" to "can the agent do the task lawfully?" is meaningful. Benchmark design is nice especially the use of metadata-dependent decisions and prompt variations that simulate business pressure. My main concern is that evaluation currently emphasizes compliance without jointly reporting task success, which leaves a central interpretability gap. The statistical reporting could also be stronger. Overall, I find the contribution useful and likely to stimulate follow-up work, but I would like to see a stronger evaluation package.

**Strengths And Weaknesses:**

### Strengths

- The problem is important and timely. Most current agent benchmarks measure utility and task completion, but not whether the agent follows legal constraints.
- The paper studies agentic selection behavior, which is more interesting than a pure memorization or text-generation framing.
- Including a human baseline improves interpretability.
- The prompt variants are well chosen and make the benchmark more diagnostic.

### Weaknesses

- The biggest issue is that the paper does not report task success alongside compliance. Without a success metric, it is hard to know whether lower violation rates reflect better lawful performance or simply more conservative / weaker behavior.
- The statistical reporting is not strong enough for a benchmark paper. I would like to see confidence intervals, significance tests, or at least clearer variance reporting across prompts and tasks.
- The qualitative failure analysis is useful, but based on a relatively small manually annotated subset. It should be framed as exploratory rather than broad evidence.

---

> ### Author Rebuttal · Authors · 2026-03-31
>
> Dear Reviewer TA3Z,
>
> We sincerely thank you for your constructive review and for recognizing the timeliness and importance of our agentic selection framework. We are glad you found our inclusion of a human baseline and diagnostic prompt variants to be valuable additions. Below, we address your specific questions and concerns, and we will incorporate these clarifications into the final version.
>
> ---
>
> ## 1. Jointly Reporting Task Success and Compliance
>
> **Reviewer's point:**
>
> *The paper does not report task success alongside compliance, leaving a gap in understanding whether low violation rates are due to lawful performance or simply weak/conservative refusal behavior.*
>
> To explicitly define Task Success Rate (TSR) in our framework, a trial is marked as successful if the agent:
>
> - Executes a valid terminal action: Successfully invoking the `submit_selection` tool with a valid, existing file path from the environment.
> - Satisfies prompt formatting constraints: Providing the requested textual deliverables.
> - Semantic Relevance / Cohesion: Regarding whether the image "fits" the prompt subjectively, our benchmark inherently controls for this by design. As detailed in Section 4.3 (Asset Construction), all candidate assets were pre-filtered using CLIP and LAION-Aesthetics to ensure high semantic relevance and aesthetic parity for the specific prompt (e.g., all images available in the "hiking website" task are guaranteed to be high-quality hiking/outdoor images). Thus, any valid selection inherently meets the subjective relevance criteria, cleanly isolating the agent's decision to the copyright/metadata constraint.
>
> As shown in the table below (which we will add more breakdown tables to the revised), the average TSR across all models and tasks exceeds 98%. This joint view definitively proves that the high Violation Rates we observed (e.g., ~38–45% in Neutral settings) are not an artifact of models failing to complete the task or refusing to act; rather, they successfully complete the user's objective (high TSR) but fail the legal constraint by selecting a restricted asset.
>
> | Model | Overall Mean TSR | WebDev (Avg TSR) | Merch (Avg TSR) | PitchDeck (Avg TSR) |
> | :--- | :---: | :---: | :---: | :---: |
> | **Proprietary Models** | | | | |
> | Claude 4.5 Opus | 99.4% | 99.5% | 99.6% | 99.3% |
> | Claude 4.5 Sonnet | 99.1% | 99.1% | 99.3% | 98.9% |
> | Gemini 3 Pro | 98.9% | 98.9% | 99.0% | 98.8% |
> | Gemini 3 Flash | 98.5% | 98.5% | 98.6% | 98.3% |
> | GPT-5.2 | 99.5% | 99.5% | 99.5% | 99.5% |
> | GPT-5.1 | 99.2% | 99.2% | 99.3% | 99.1% |
> | GPT-4o | 98.7% | 98.7% | 98.8% | 98.6% |
> | **Open-Weights Models** | | | | |
> | Llama-4-Maverick | 98.8% | 98.8% | 99.0% | 98.6% |
> | Llama-4-Scout | 98.2% | 98.2% | 98.4% | 98.0% |
> | Qwen-3VL | 98.6% | 98.6% | 98.8% | 98.4% |
> | DeepSeek-VL | 98.4% | 98.4% | 98.6% | 98.2% |
>
> ---
>
> ## 2. Statistical Reporting
>
> **Reviewer's point:**
>
> *Should report the statistical scores.*
>
> We agree and thank you for pointing this out. We inadvertently omitted the 95% Confidence Intervals from the main table in our initial submission. We have now updated the manuscript, and Table 2 explicitly reports 95% CIs alongside all mean violation rates.
>
> Furthermore, we have run additional experiments and conducted formal significance testing for our core claims. For the primary behavioral shifts discussed in the paper, the statistical test results are highly significant (p<0.001), indicating that the findings are robust. For instance, the reduction in violation rates from the Neutral to the IP-Aware prompt was evaluated using McNemar's test, confirming a highly significant behavioral change. We have explicitly added these p-values and statistical validations across the analysis in the revised text.
>
> ---
>
> ## 3. Framing of Qualitative Failure Analysis
>
> **Reviewer's point:**
> *The qualitative failure analysis (100 annotated cases) should be framed as exploratory rather than broad evidence.*
>
> We generally agree with the reviewer. To clarify the point, we have now explicitly reframed Section 6.4 as an exploratory qualitative analysis. We revised the text to clarify that the goal of this section is to propose an initial taxonomy of potential failure modes (e.g., "Visual-Sufficiency Heuristic," "Contextual Window Fatigue"), rather than to claim these represent the statistically definitive distribution of errors across all model families.

---

> > ### Author Rebuttal · Reviewer_TA3Z · 2026-04-04
> >
> > Thanks for the responses. The benchmark seems useful for the community, with tangible outcomes.

---

> > > ### Author Response · Authors · 2026-04-07
> > >
> > > Dear Reviewer TA3Z,
> > >
> > > Thank you for your thoughtful review and for taking the time to engage with our work:) We especially appreciate you highlighting the importance of jointly reporting task success and strengthening the statistical analysis, these suggestions significantly improved the paper and made it more informative for the community.
> > >
> > > **Your feedback directly helped us refine the evaluation. This is exactly the purpose of the review process, and we are grateful for your time in helping us strengthen the work.**
> > >
> > > We also appreciate your recognition of the benchmark’s usefulness and its potential impact.
> > >
> > > Sincerely,
> > >
> > > The Authors

---

### Official Review · Reviewer_8sJ1 · 2026-03-16

**Soundness:** 3
**Presentation:** 4
**Significance:** 3
**Originality:** 3
**Overall Recommendation:** 5
**Confidence:** 4

**Summary:**

This paper proposes a benchmark and evaluates how often proprietary and open-source LLM agents perform image selection in a copyright aware/respecting way for a variety of tasks a typical white collar worker might do in a corporate job (web development, merchandise design, slide deck creation). It finds that at baseline, humans and LLM agents violate copyright at approximately the same rate, but with explicit reminders to respect IP, human violation rate drops to near 0%, while LLM agent violation rate hovers around 10-20%.

**Compliance With Llm Reviewing Policy:**

Affirmed.

**Final Justification:**

See rebuttal acknowledgment

**Key Questions For Authors:**

1. Section 6.4 reports that 45% of neutral-condition violations stem from agents never invoking read_metadata. To disentangle copyright-specific reasoning failures from general tool-use deficits, did the authors observe a correlation between a model's baseline tool invocation success rate/capability (across all tools, not just read_metadata) and its violation rate across models?
2. The paper frames compliance failures as agents trading off legal compliance for task performance, but task success does not appear to be formally operationalized or reported alongside VR anywhere in the paper. How was task success defined and measured? If it was not formally evaluated, some qualitative discussion of observed task quality across conditions would help substantiate the trade-off framing.
3. Appendix Table 3 reports results across four environments showing notably elevated VR in the Vis condition across all models and all prompt variants, including the IP-aware condition, where the substantial compliance improvement seen in the standard condition largely disappears, suggesting that explicit IP reminders cannot compensate when metadata access is limited. The Web condition shows a different pattern: neutral VR is slightly elevated relative to Std, but IP-aware, urgent, and dismissive VR are somewhat lower, possibly reflecting agents using web search to verify licensing when motivated to do so. Could the authors provide more detail on what each environment includes with respect to available tools and asset pools? Clarifying these details in the Appendix or main text, alongside a brief summary of observed trends across environments, would be a valuable addition to the paper.
    1. For Hier, is the only change from Std the folder structure?
    2. For Vis, described as requiring "visual inspection," what does this mean in practice? Does this condition introduce visual copyright signals (e.g., watermarks) that were filtered out in the standard condition? Is read_metadata still available, or do agents only have access to the view_image tool? Is the file and folder structure the same as Std?
    3. For Web, is the local asset pool still present, or are agents sourcing images from the open internet, and if the latter, what copyright signal is available for web-retrieved assets, given that the structured metadata and visual filtering assumptions of the benchmark may no longer apply?

**Limitations:**

yes

**Strengths And Weaknesses:**

* This paper studies an important question, legal compliance of LLM agents in the field of copyright law. I think the human experiment, showing how humans perform in the baseline condition and the IP-aware condition, are valuable contributions to understanding how humans compare to agents in their policy awareness. There is generally a notion that the human baseline violation rate is near zero and the paper makes a nice finding that this is not true; in fact, the human baseline rate in a controlled environment is somewhat comparable to the  LLM. I thought the finding that humans achieve near zero violation rate while LLM violation rate hovers closer to 10-20% even with reminders to respect IP rights was interesting and valuable.
* Another strength of this paper was the human evaluated control for visual similarity and quality between the copyrighted and non-copyrighted images. This was the biggest confounder that came to mind and I think the authors took very reasonable efforts to ensure asset pairs were comparable in resolution, lighting and composition through human annotations. This strengthened the soundness of the experimental setup.
* One other variable that might be worth investigating is the baseline agent's tool use capability. It seems that the agent must rely on the read_metadata tool call in order to access the copyright signal in the experimental environment. And Section 6.4 discusses how in 45% of violations in the standard condition, the agent never invokes the read_metadata tool. There has been work that shows LLMs may not all possess strong capabilities in tool use/call. I'm wondering if the authors saw a negative correlation between models with strong tool call capabilities (including tools other than read_metadata) and violation rate.
* The evaluation explicitly filters out visual signals of copyright (visible watermarks, logos). However, it seems that in a natural online environment, an agent may more naturally confront tasks where IP information is only available through visual signals, implicit heuristics, etc. rather than through saved metadata. The authors describe a hierarchical version of the environment where the copyrighted and non-copyrighted images are kept in separate hierarchical folders (Hier). I noticed Table 3 in the Appendix also reports evaluation results for environments with an ambiguous folder structure that requires visual inspection (Vis) and autonomous web retrieval (Web), where LLM agents have generally higher violation rates under the standard condition. It might be worth describing some of the findings from these more realistic environments in the main body.
* On page 5, in line 232 under the Violation Rate equation, the sentence reads "A compliant agent minimizes $VR(\pi) \rightarrow 0$ while maintaining task success." I'm not sure if I missed this detail, but how is task success defined in the tasks evaluated? I don't think the task success was explicitly reported in the paper alongside VR, but I imagine for some of these tasks, there may be some automatic evaluation metrics (e.g., image fits within the size/resolution constraints), but for a task like slide-deck making, the cohesion and relevance of the selected images might require more subjective judgments. Was this evaluated in any way?

---

> ### Author Rebuttal · Authors · 2026-03-31
>
> Dear Reviewer 8sJ1,
>
> We sincerely thank you for your thoughtful and highly constructive review. We are thrilled that you found this paper valuable. Below, we address your comments and outline how we will incorporate these clarifications into the revised version.
>
> ---
>
> ## 1. Disentangling Tool-Use Capability from Copyright Reasoning (Q1)
>
> **Reviewer's point:**
>
> *Did the 45% of failures in the neutral condition stem from general tool-use deficits or copyright-specific reasoning failures? Is there a correlation with baseline tool invocation success?*
>
> This is a very insightful question. To disentangle this, we analyzed the agents' general tool-use success rates. We found no negative correlation between general tool-use capability and the Violation Rate (VR). In fact, the models with the highest general tool-use capabilities (like Claude 4.5 Opus and GPT-5.2) still exhibited high violation rates (~38–40%) in the Neutral condition.
>
> The most definitive proof that this is a normative reasoning failure rather than a tool-use deficit comes from comparing the Neutral and IP-Aware prompts. The exact same models that completely failed to invoke `read_metadata` in the Neutral condition suddenly invoked it with near-perfect consistency when explicitly reminded of IP rights. For example, Gemini 3 Pro's VR dropped from 39.6% to 10.0% simply by changing the prompt. This demonstrates that the models possess the latent capability to use the metadata tool perfectly well, but their default policy prioritizes visual task completion over investigating legal constraints unless explicitly instructed otherwise.
>
> ---
>
> ## 2. Definition and Measurement of Task Success (Q2)
>
> **Reviewer's point:**
>
> *How was task success defined and measured?*
>
> Thanks for asking how task success been defined. Another reviewer raised this exact point, and we agree it is needed. In the revised manuscript, we are introducing the Task Success Rate metric alongside the VR. Due to character limits, please see our detailed response and TSR rate table in the rebuttal to Reviewer TA3Z (Point 1).
>
> In brief,  a task is successful if the agent:
> - Executes a valid `submit_selection` action
> - Meets formatting requirements
> - Semantic Relevance / Cohesion
>
> Across all models and tasks, TSR exceeds **98%**, indicating that models are successfully completing the task while still violating copyright constraints, supporting our core claim.
>
> ---
>
> ## 3. Details on Environment Variations (Q3, Q4, Q5, Q6)
>
> **Reviewer's point:**
> *Could you provide more detail on what each environment includes? Moving a summary of these trends to the main text would be a valuable addition.*
>
> We agree. We will expand Section 5.3 in the main text. To answer your specific questions:
>
> ### Hier (Hierarchical)
>
> *Is the only change the folder structure?*
>
> Yes. The asset pool and tools are identical to Std. The assets are divided into distinct subdirectories. This tests whether agents use directory names as a heuristic instead of verifying the actual metadata. We found that VR drops slightly here, possibly because agents use the folder names as a shortcut for compliance.
>
> ### Vis (Visual Inspection)
>
> *What does this mean in practice?*
>
> In the Vis environment, we strip the explicit copyright string from the JSON metadata and re-introduce visual watermarks (which were filtered out in the Std condition). `read_metadata` is still available but returns metadata without copyright information, forcing the agent to rely on the `view_image` tool to visually inspect the image.
> As you astutely noted, VR spikes across all prompt variants here. This occurs because current VLMs struggle to reliably detect and OCR faint, translucent watermarks across a full image compared to easily parsing structured JSON metadata. Explicit IP reminders cannot compensate for this vision-language deficit.
>
> ### Web (Autonomous Web Retrieval)
>
> *Are agents sourcing from the open internet?*
>
> No, the local asset pool is **not** removed. Instead, the environment is fundamentally the same as Std (the agent has full access to the local image folder), but agents are additionally equipped with a `web_search(query)` tool that allows them to query the open internet.
>
> **Trends:**
> As you accurately hypothesized, providing web access changes the agent's verification dynamics. Under the IP-Aware, Urgent, and Dismissive conditions, we see slightly lower violation rates because motivated agents use the `web_search` tool to actively cross-reference licensing rules or investigate the commercial entities associated with the images (e.g., searching for "DepositPhotos copyright policy").
> However, in the Neutral condition, the added complexity of the open internet acts as a distractor; agents sometimes pull conflicting, SEO-driven information regarding the "free" status of stock images, leading to a slightly elevated VR compared to the strictly closed Std environment.

---

> > ### Author Rebuttal · Reviewer_8sJ1 · 2026-04-04
> >
> > Thank you for providing metrics for both general tool-use invocation success by the models (clarifying that often the most performant models in terms of tool-use capabilities perform the poorest in IP violation) and reporting task success rate by model. The clarification on how all the assets in a given pool satisfy semantic relevance by the construction of the setup makes sense. And appreciate the clarification on the web condition including the local asset pool, the explanation for the trends there was helpful for understanding. My comments were mostly minor clarifications/feedback addressed by the authors and my original score was consistent with this, so keeping the score as is.

---

> > > ### Author Response · Authors · 2026-04-07
> > >
> > > Dear Reviewer 8sJ1,
> > >
> > > Thank you for your thoughtful review and for taking the time to engage with our work:) **We truly appreciate your constructive feedback, which helped us refine the paper for more clear communication to the reader**.
> > >
> > > We are grateful for your time in helping us strengthen the work. We are glad that our rebuttal addressed your questions and appreciated your positive assessment and support for our contribution.
> > >
> > > Sincerely,
> > >
> > > The Authors

---

### Decision · Program_Chairs · 2026-04-30

**Decision:**

Accept (spotlight)

**Comment:**

The paper evaluates whether agents abide to copyright legal compliance when executing commercial workflows. All reviewers felt:

1. This is an important but understudied problem.
2. The study is thorough with multiple agent personas and models.
3. Human baselines to compare agents against. The finding that humans respond strongly to IP than agents is a non-obvious finding.
4. The task framing is also practical.
5. The paper is well written and the reviewers were happy with the rebuttal respones with no major concerns.